

# Brief Communication: 3D landslide motion from cross correlation of UAV-derived morphological attributes

Maria V. Peppa[1], Jon P. Mills[1], Phil Moore[1], Pauline E. Miller[2], Jonathan E. Chambers[3]

[1]School of Civil Engineering and Geosciences, Newcastle University, Newcastle upon Tyne, UK
[2]The James Hutton Institute, Aberdeen, UK
[3]British Geological Survey, Keyworth Nottingham, UK

*Correspondence to*: Maria V. Peppa (m.v.peppa@ncl.ac.uk)

**Abstract.** Unmanned aerial vehicles (UAVs) can provide observations of high spatio-temporal resolution to enable operational landslide monitoring. In this research, the construction of digital elevation models (DEMs) and orthomosaics from UAV
imagery is achieved using structure-from-motion (SfM) photogrammetric procedures. The study examines the additional value that morphological attribute of *openness*, amongst others, can provide to surface deformation analysis. Image cross-correlation functions and DEM subtraction techniques are applied to the SfM outputs. Through the proposed integrated analysis, the effective, automated quantification of a landslide's 3D motion over time is demonstrated, with implications for the wider interpretation of landslide kinematics via UAV surveys.

**1 Introduction**

Landslides are a form of mass movement, which can often be complex in nature, leading to slope failure and the formation of surface morphological structures. Monitoring of these structures can provide a valuable insight into a landslide's sub-surface dynamic failure mechanism and thereby help mitigate hazards (Gunn et al., 2013). Conventionally, in addition to geotechnical and geophysical monitoring of the sub-surface, survey markers are often used to quantify surface displacement by monitoring
discrete locations through periodic observations. However, such surveying can be hazardous and generally provides limited spatial resolution. The development of low cost, mini consumer-grade UAVs – also known as remotely piloted aircraft systems (RPAS), and drones –  equipped with off-the-shelf compact cameras, in combination with structure-from-motion (SfM) and multi-view stereo (MVS) algorithms, has facilitated high spatio-temporal resolution topographic surveys using image-based approaches. In particular, the implementation of the SfM-MVS pipeline into user-friendly commercial software packages, such
as PhotoScan (PhotoScan, 2016) and Pix4D (Pix4D, 2016), has enabled the generation of high spatio-temporal resolution point clouds, DEMs and orthomosaics in the Earth sciences (Remondino et al., 2014; James et al., 2017).

Differencing of successive co-registered DEMs constitutes a standard approach to estimate ground accumulation and depletion in monitoring applications (Daehne and Corsini, 2013; Travelletti et al., 2014). Moreover, image cross-correlation functions applied to optical imagery has long been successfully implemented for the quantification of surface planimetric movement in
the context of landslides, glaciers, etc. (Leprince et al., 2007; Ayoub et al., 2009a; Heid and Kääb, 2012). Nevertheless, the





application of image cross-correlation functions to UAV-derived orthomosaics can increase noise due to variations in illumination conditions (Lucieer et al., 2014). Recent studies have demonstrated that the implementation of image cross-correlation functions with DEM morphological derivatives can automatically determine the movement of surface features that preserve their structural patterns over time (Daehne and Corsini, 2013; Lucieer et al., 2014; Travelletti et al., 2014; Fey et al.,

2015). Among these, Lucieer et al. (2014) and Turner et al. (2015) found the UAV-derived morphological attribute of *shaded relief,* implemented with image cross-correlation functions, to provide better surface displacement estimation of a landslide than single bands from the corresponding orthomosaic. To date, however, there has been no comprehensive evaluation of image cross-correlation functions with various UAV-derived morphological attributes for 3D landslide deformation monitoring.

This paper reports on the analysis of 3D surface change of an active landslide from multi-temporal co-registered UAV-derived outputs, including DEMs, orthomosaics, and morphological attributes. Firstly, image cross-correlation functions are evaluated through comparative analysis with synthetic datasets. Secondly, the 3D surface deformation of a landslide is determined by integrating image cross-correlation functions with morphological attributes and DEM differencing. The paper illustrates how to exploit a time-series of UAV survey derivatives in order to quantify and interpret landslide kinematics.

## 2 Study area

The Hollin Hill study site comprises a slow moving earth-slide, earth-flow landslide with an extent of 290 m E-W, 230 m N-S and a south-facing slope of an average 12º, located in the Lias mudrocks of North Yorkshire, UK (54º 6' 38.90" N, 0º 57' 36.84" W). The site has been monitored since 2009 by British Geological Survey (BGS) using various methods, including terrestrial and airborne laser scanning, as well as ground-based geotechnical and geophysical investigations. BGS

investigations have revealed that the landslide has a complex behaviour with seasonal surface variations and episodic failures mostly triggered by intensive rainfall and increased pore-water pressures within the constituent geological materials (Gunn et al., 2013; Uhlemann et al., 2017).

## 3 Data acquisition and processing

Image acquisition was performed using a mini fixed-wing UAV (Quest 300) equipped with a Panasonic Lumix DMC-LX5

compact camera of 5.1 mm nominal focal length and an image array of 3648 x 2736 pixels. RGB UAV imagery was captured during six field campaigns in December 2014, March 2015, June 2015, September 2015, February 2016 and May 2016. The Quest 300 was flown from a nominal flying height of 90 m at 18 m/s, with images acquired approximately every 2 s. During every field campaign, a GNSS base station was established over stable terrain and surveyed in static GNSS mode. Average absolute accuracies of 0.01 m in planimetry and 0.02 m in elevation were delivered. Circular targets, for which centres could


be easily recognisable on the images, were established and surveyed in rapid static Global Navigation Satellite System (GNSS) mode with between 11 and 20 targets for each of the different campaigns.

A self-calibrating bundle adjustment, incorporated into the SfM-MVS pipeline, was utilised to process the UAV imagery using PhotoScan software, as described in Peppa et al. (2016). The observed coordinates of five circular targets were utilised as

control in each SfM-MVS bundle adjustment, with the remainder used as independent check points. This resulted in the reconstruction of six dense point clouds, one per epoch, georeferenced in the Ordnance Survey Great Britain 1936 (OSGB36) coordinate system. From an average 0.03 m ground sample distance, DEMs were generated at each epoch with an average 0.06 m spatial resolution. The 3D co-registration accuracy, calculated from differences between the surveyed and observed coordinates at independent check points after the SfM-MVS bundle, was estimated as an average root mean square error

(RMSE) of 0.03 m. An approximate ±0.10 m sensitivity level, corresponding to the lowest detectable 3D change, was estimated by applying error propagation to the 3D RMSE values with a 95% confidence level.

## 4 Methodology

Four morphological attributes (*shaded relief, slope, openness* and *curvature*) were computed from each epoch's DEM. *Shaded relief* was created with the aid of the ambient occlusion tool in the SAGA GIS package. This applies homogenous illumination

to the DEM, smoothing the shadow effect usually produced by lighting from a single direction (Fey et al., 2015). The remaining three morphological attributes were all generated using the Orientation and Processing of Airborne Laser Scanning data (OPALS) software (Pfeifer et al., 2014). In this paper: a) *slope* indicates the steepest slope angle of the surface; b) *openness* represents the minimum angle of a cone fitted in the DEM, as viewed from above the surface (Yokoyama et al., 2002); c) *curvature* constitutes the average of minimum and maximum curvature, representing concave and convex surface features

respectively. All three attributes were computed using a 3x3 pixel radial distance, equivalent to 0.18 m at 0.06 m pixel resolution.

An experiment was conducted with synthetic epoch pairs to evaluate the performance of the statistical normalised cross-correlation (NCC) function, implemented in the Co-registration of Optically Sensed Images and Correlation (COSI-Corr) software (Leprince et al., 2007; Ayoub et al., 2009b), as applied to these four morphological attributes. To generate the

25 synthetic displacement, known translations of a) 0.050 m in Easting and -0.100 m in Northing (i.e. 0.112 m total magnitude) were applied to Region A (see Figure 1a), approximating to the ±0.10 m sensitivity level; and b) shifts of 0.455 m in Easting and -0.544 m in Northing (0.709 m total magnitude) applied to Region B in the December 2014 DEM, simulating typical inter-epoch movement of the real landslide. Four pairs of morphological attributes were then derived from both the original December 2014 DEM and the synthetically shifted DEM. Each pair, comprising the pre- and post-event images, was imported

into the COSI-Corr function. This computes the maximum absolute value of the correlation coefficient by sliding a rectangular patch from the pre-event image systematically within a window in the post-event image. The computed displacements in Easting and Northing, determined by the matched correlation peak between the two images, have a spatial resolution equal to





a specified step parameter used for the sliding (Ayoub et al., 2009b; Lucieer et al., 2014). After a trial and error procedure, a window size of 64x64 pixels (3.84 m) with a step of 16x16 pixels (0.96 m) and a patch of 20x20 pixels (1.20 m) were chosen for this research. These settings ensured that the maximum imposed shift over Region B could be detected and was therefore chosen in line with a priori knowledge of the Hollin Hill landslide movement rates (Uhlemann et al., 2017).

Apart from the displacements in Easting and Northing, the COSI-Corr function also calculates a signal-to-noise ratio (SNR), indicative of the correlation quality. SNR values closer to unity are indicative of more reliable results. A comparative analysis of the estimated displacements and derived SNRs, obtained with the four morphological attributes, was then performed to determine which of the morphological attributes produced the optimal results. The chosen morphological attribute, together with the COSI-Corr function, was applied to successive epoch pairs of the Hollin Hill landslide to estimate 2D motion. The

COSI-Corr result was cross-validated with the surface displacements calculated from 27 sample points, manually measured across the orthomosaics. Elevation change was derived by subtracting each DEM from the subsequent DEM on a pixel-by pixel basis.

Having generated a time-series of 3D surface deformation across the site, an additional investigation over sub-regions with the largest deformations was then performed. The morphological attribute of *openness* was then chosen due to its unique

representation of discernible surface patterns within the landslide body. The NCC function was applied to *openness* for December 2014 and May 2016 datasets, as implemented in the Correlation Image Analysis (CIAS) package (CIAS, 2012; Heid and Kääb, 2012) using the aforementioned window and patch sizes. Unlike COSI-Corr, CIAS allows individual feature tracking. Thus, characteristic surface structures were manually located over the December 2014 *openness* image and the total planimetric vectors were automatically derived with the CIAS tool. Manual cleaning to remove spurious vectors was also

necessary, although this process could be automated by application of various threshold parameters, if necessary.

**5 Results**

Regarding the synthetic experiment, all four morphological attributes underestimated the imposed displacement of Region A, delivering an average displacement 0.030 m ± 0.027 m in Easting and 0.054 m ± 0.030 m in Northing. For Region B, the closest result to truth in Easting was delivered by *openness*, with an average value of 0.435 m ± 0.145 m, whereas *shaded relief*

detected the best average displacement in Northing of -0.528 m ± 0.131 m. Figure 1a and 1b depict the SNR results, derived from *openness* and *shaded relief* respectively, over stable terrain outside Regions A and B. Figure 1c presents the boxplots of the comparative SNR analysis. SNR values close to zero (Figure 1b) indicated decorrelation, which is also illustrated as outliers in the boxplot of *shaded relief* over stable terrain, whereas the other three morphological attributes were less noisy (Figure 1c). For Regions A and B all morphological attributes with the exception of *curvature* produced similar boxplots. The boxplots

reveal greater variation in SNR in Region B than in Region A (Figure 1c), possibly due to the noise caused by the extreme local surface variations around Region B. Overall, *slope* and *openness* provided comparable displacements and noise levels.





In this study, *openness* was finally chosen for the estimation of Hollin Hill landslide motions, as it highlights characteristic breaks in slope sliding downwards over time.

The comparison of the COSI-Corr derived-displacements with the manually observed surface movements at 27 sample points (Figure 2) indicates the sensitivity of the NCC function to different displacement magnitudes. The scatterplot in Figure 2 shows a general systematic overestimation of the displacement magnitude derived from COSI-Corr. Some scattered points fell within the ±0.10 m 3D sensitivity level shown in grey. Significant movement was observed mostly between December 2014-March 2015, September 2015-February 2016 and February 2016-May 2016 epoch pairs. Overall, the NCC function delivered results in good agreement with the manual measurements (closer to the straight line) for small displacements, but miscalculated the surface movement of the last epoch pair.

The planimetric displacements across the Hollin Hill landslide between December 2014-March 2015, March 2015-February 2016 and February 2016-May 2016 are mapped in Figure 3a, Figure 3b and Figure 3c respectively. Observations from June 2015 and September 2015 campaigns were excluded from the maps in Figure 3 due to small displacements and additional noise caused by vegetation change. Blue hatched polygons represent areas with more reliably estimated surface displacements, as the SNR is greater than 0.7. This value is equivalent to the lowest whisker of the *openness* boxplot (Figure 1c), representing the outlier threshold, as derived from Eq. (1):

$$lowest\ whisker = Q1 - 1.5x(Q3 - Q1) \tag{1}$$

where $Q1$ and $Q3$ the 25% and 75% percentiles of the data respectively.

There are a few erroneous displacements, mostly at the edges of the study site, around vegetated areas and outside the blue hatched polygons, as evidenced in Figure 3a, Figure 3b and Figure 3c. The elevation differences between the same epoch pairs are depicted in Figure 3d, Figure 3e and Figure 3f, excluding deformations within the ±0.10 m sensitivity level. For instance, Figure 3f depicts the grass growth at the foot of the slope, which in turn caused false surface movement in Figure 3c. Also, over the regions with extreme deformations (e.g. back scarp in Figure 3c) decorrelation was generated creating voids on the displacement map.

To further investigate these significant deformations, the May 2016 *openness* image was superimposed over the corresponding image from December 2014 and is presented in Figure 4a and 4b. Figure 4c illustrates that narrow angles of *openness* can distinguish surface undulations sliding down-slope. For instance, point 1 moved 1.10 m along the profile AB towards the south. To visualise these structures a threshold of 63° was applied to the *openness* images (Figure 4a and 4b). Different thresholds can visualise different morphological features. *Openness* also captured the surface rupture that occurred at the top of the slope between February and May 2016 (Figure 3f and 4b). The planimetric vectors of distinctive features are plotted in Figure 4a and 4b, as automatically determined after applying the NCC function implemented in CIAS. Spurious vectors at the edges of the back scarp, which were manually removed, were possibly generated due to rotational failures investigated by BGS (Uhlemann et al., 2017).





## 6 Discussion

The comparative analysis of the NCC function with synthetic data was necessary to tune the function's optimal settings. If small displacements close to the UAV-derived sensitivity level do not fit within the specified window size, they cannot be precisely estimated (e.g. Region A), as was noted by Fey et al. (2015). Small step and window sizes improved the spatial

resolution of the surface displacement magnitude map but increased the computational time and noise. This occurred as features with similar / repetitive patterns within the vicinity of the specified window sizes generated false displacements (Travelletti et al., 2014; Fey et al., 2015). Hence, the choice of the function's parameters is usually based on the required spatial resolution, the computational effort and the displacement magnitude (Daehne and Corsini, 2013; Travelletti et al., 2014; Fey et al., 2015).

The analysis with synthetic data also demonstrated that imagery derived from various morphological attributes can generate different displacement estimations and noise levels. *Slope*, *openness* and *curvature* outperformed *shaded relief* in terms of noise over stable terrain, even though all attributes are insensitive to illumination variations and shadows (Daehne and Corsini, 2013; Lucieer et al., 2014; Fey et al., 2015). A possible error source could be the grass cover, well known to affect the results of image cross-correlation (Lucieer et al., 2014; Stumpf et al., 2017).

The production of reliable surface displacements with the image cross correlation functions over vegetated terrain constitutes a significant challenge. As vegetation covers surface features, the NCC function generates additional noise. Conversely, grassy surfaces produce images with low texture and without distinctive surface features which can also affect the NCC function's performance (Travelletti et al., 2014), as evidenced in Figure 1b, Figure 3a, 3b and 3c around the eastern lobe. Hence, noisy results attributed to vegetation presence cannot be entirely removed, even with UAV surveys of high temporal resolution. The

use of morphological attributes computed with larger spatial distances, thereby producing a higher level of smoothing, can potentially decrease this noise. Finally, to generate optimal NCC results with the least amount of noise possible, accounting for the vegetation variation, winter would constitute the best period to conduct UAV surveys.

Independently of the NCC function's sensitivity to displacement magnitude and vegetation presence, the presented analysis also revealed other limitations, already well reported in previous studies (Daehne and Corsini, 2013; Lucieer et al., 2014;

Travelletti et al., 2014; Fey et al., 2015; Stumpf et al., 2017). A priori knowledge of the displacement magnitude is required for tuning the function's settings, therefore somewhat limiting the automated fashion of the workflow. In addition, decorrelation occurs when a surface has significantly changed between two consecutive epochs. Finally, image cross-correlation functions generate unreliable estimations over regions with rotational failures, creating spurious vectors or voids, whereas performance is much better over translational earth flow slides.

Overall, heterogeneous 3D surface changes were observed at the Hollin Hill landslide through the combination of multiple co-registered UAV products. The orthomosaics supported the identification of vegetated areas and cross-validation of the results. The use of *openness*, together with the COSI-Corr tool supported the quantification of the movement over the whole site. DEM differencing was also applied to quantify the episodic surface ruptures and interpret the generated voids on displacement maps.





The CIAS tool applied to *openness* tracked the evolution of discernible surface patterns over the eight-month duration in a semi-automated fashion. *Openness* maps of different angle thresholds express surface formations in different ways, and as a result can complement investigation of landslide motion. The exploitation of available image cross-correlation tools (COSI-Corr and CIAS) with *openness* decreased the intensive task of manual feature tracking. However, this task is still essential for cross-validation, especially in cases where ground truth observations are lacking over the monitoring period.

# 7 Conclusions and future work

This paper has presented an investigation of UAV-derived products of DEMs and orthomosaics along with DEM morphological derivatives of *openness* to automatically quantify the spatio-temporal motion of an active landslide. The research has demonstrated the successful integration of image cross-correlation functions with morphological attributes and the importance of the comparative analysis with synthetic data. The analysis has illustrated that the fusion of *openness* morphological attribute along with DEM differencing can support the comprehensive interpretation of landslide behaviour, providing a holistic overview of 3D surface deformation patterns. Future work will assess the performance of image cross-correlation functions with *shaded relief, slope, curvature* and other possible DEM derivatives, computed with various pixel radial distances, implemented with real-world data. It will also apply other techniques to automatically filter spurious results. Ultimately, future research will investigate the correlation of the 3D motion time-series with rainfall observations to enhance the understanding of the landslide mechanisms.

**Acknowledgements**

This research was jointly funded by a Natural Environment Research Council (NERC) BGS BUFI award (S241) and an Engineering and Physical Sciences Research Council (EPSRC) DTA award (EP/L504828/1) at Newcastle University, UK. The BGS contribution to this paper is published with the permission of the Executive Director of the British Geological Survey (NERC). The authors wish to acknowledge the Hollin Hill site landowners for allowing access to their property. Finally, many thanks to Martin Robertson, Elias Berra, Magdalena Smigaj, Polpreecha Chidburee and Ben Grayson, all of Newcastle University, for fieldwork assistance.

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





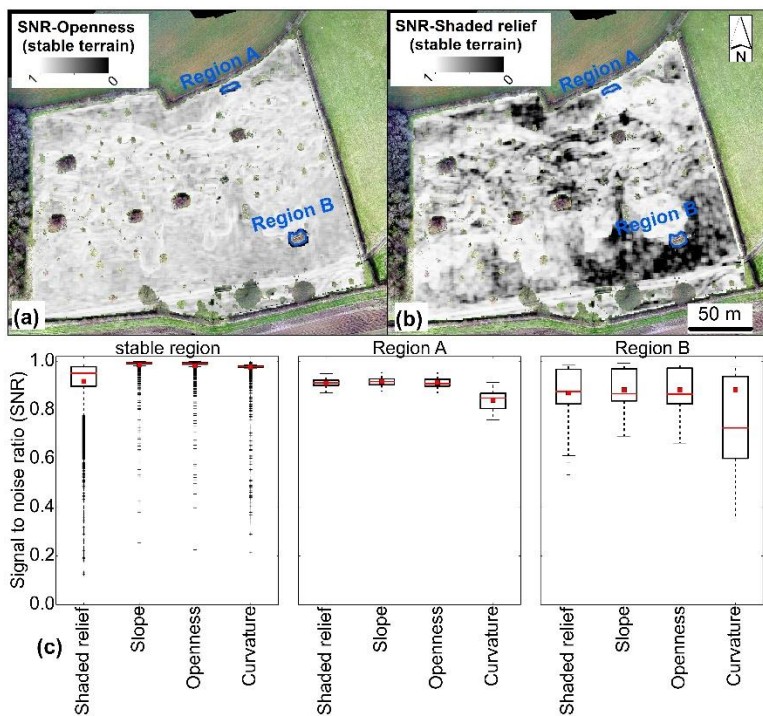

**Figure 1: SNR maps of stable terrain derived from COSI-Corr with (a)** *openness* **and (b)** *shaded relief* **superimposed over December 2014 orthomosaic. (c) Box plots of SNRs for stable terrain in Regions A and B, as derived from the implementation of COSI-Corr with** *shaded relief*, *slope*, *openness* **and** *curvature* **applied to synthetic datasets. The median is displayed as a red line, the mean as a red rectangle, the whiskers as black horizontal lines and the outliers as black crosses.**





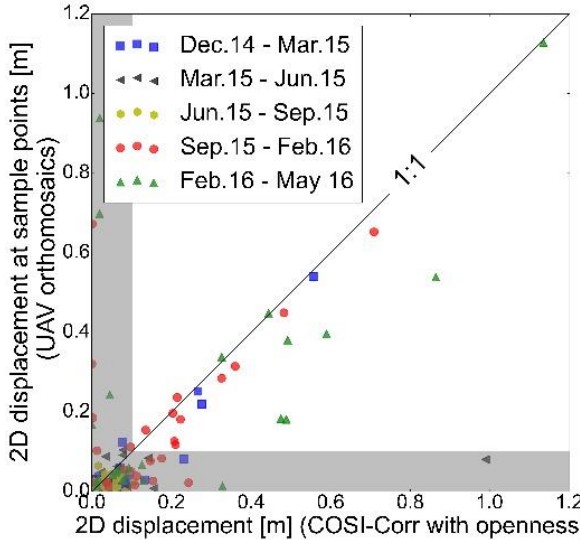

**Figure 2: Scatterplot of estimated surface displacements determined by COSI-Corr with *openness* plotted against manual observation per epoch pair.**



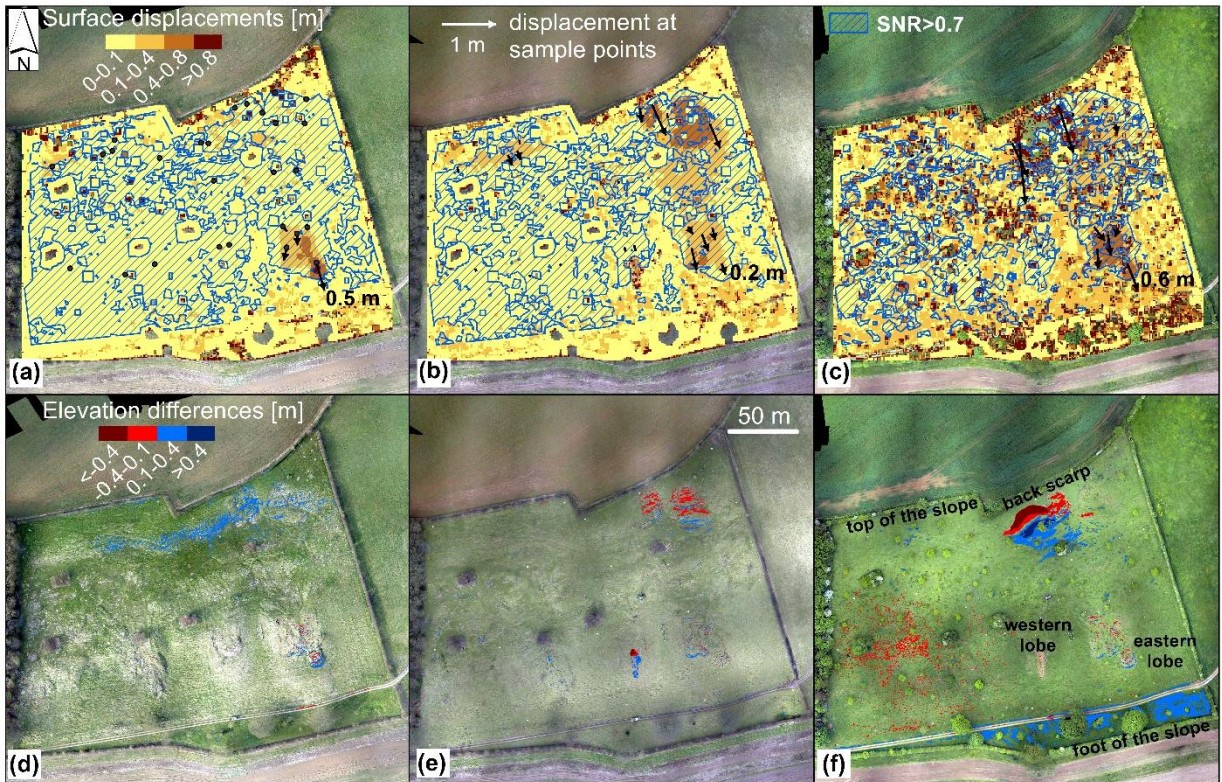

**Figure 3: Maps of surface displacements and elevation differences of (a and d) December 2014 -March 2015, (b and e) March 2015-February 2016 and (c and f) February 2016-May 2016, respectively. Manually derived planimetric vectors at sample points are also superimposed.**





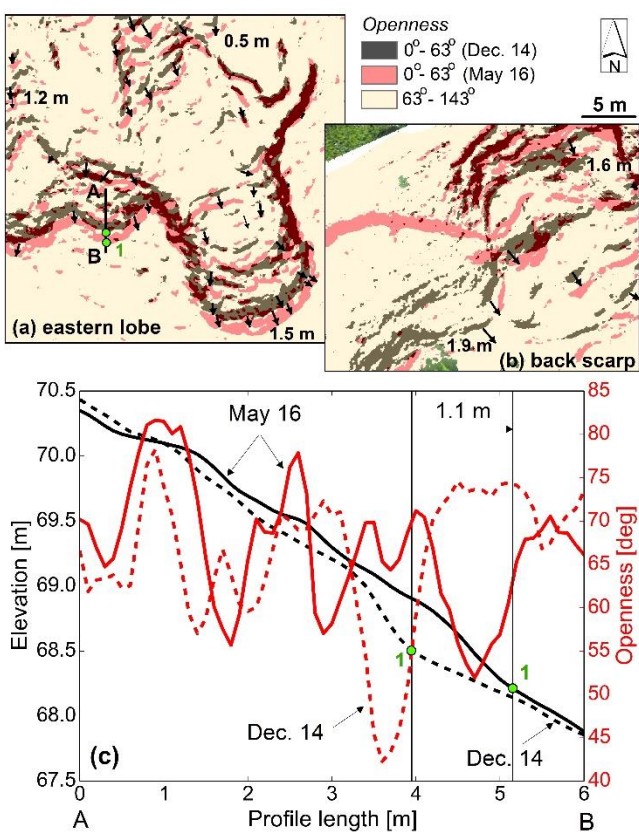

**Figure 4: Detailed view of December 2014 and May 2016 openness maps over (a) eastern lobe and (b) back scarp with elevation and *openness* plotted along (c) Profile AB.**