# Peer review of "Brief Communication: Landslide motion from cross correlation of UAV-derived morphological attributes"

_Natural Hazards and Earth System Sciences, 2017_

## Referee Comment (RC1) · Anonymous Referee #1 · 31 Aug 2017

The manuscript presents an interesting application of UAV products for remote monitoring of landslide motion. In particular, the Authors calculated DEM and derived morphological attritutes such as slope, curvature, shaded relief and opennes at six different times. Differences of DEM and morphological attributes were calculated and utilized as inputs in an existing software package to calculate cross-correlations. The procedure adopted in this work was first applied to a synthetic displacement in two test areas, simulating the motion of one real landslide, to calibrate various software parameters. Eventually, the procedure is validated against ground truth measured independently in the field. The validation procedure is apparently essential to obtain the final results: a limitation which is actually acknowledged by the Authors.
The novelty of the approach is represented by the use of many morphometric attributes, at variance with previous approaches using the sole shaded relief. The calculation of DEM and morphological attributes from UAV imagery allows fast, cheap and, most importantly, repeated in time measurements, a fundamental requirement for monitoring slow landslides.

In my opinion, the Manuscript suffers from two limitations. Though the presentation is generally clear, a few details of the method are not. Also, the conclusions drawn by the Authors are somewhat incomplete and fail to report the major criticality of the method, namely the need of calibration/validation field data.

As a matter of fact, the use of morphological attributes, their signal-to-noise ratio, the optimal choice of the best attribute and reasons for that, the possibility of describing different details of the landslide body with different thresholds for openness are described at length, while the use of DEM differencing is barely mentioned in a few points throughout the paper. This does not appear enough to understand how the technique was useful to obtain the results. Both the Abstract and the Conclusions section explicitly mention that "the analysis has illustrated that the fusion of openness morphological attribute along with DEM differencing can support the comprehensive interpretation of landslide behaviour", implying that the technique was equally important than the morphological analysis thus the Authors should describe how they performed the "fusion".

Another unclear point, though less important, is the description of manual removal of spurious displacement vectors. How did the Authors distinguish such vectors? Is the sentence at lines 27-29, page 6, about rotational features enough to describe that, and is the knowledge of presence of rotational (local?) failures a further requirement for the successful application of the procedure? Is it so straightworward to define proper thresholds to automatically remove the spurious vectors, or does it require further calibration, thus relying even more on field data?

I also believe that the lines 21-31, page 4, and 1-2, page 5, in the Results section,
belong to the Method section. In fact, they characterize the "calibration" step, so that they are not results of the method, strictly speaking.

In line 22, page 6, it was suggested that "winter would constitute the best period to conduct UAV surveys", due to presence of vegetation constituting a major noise source. Is it possible to deduce this statement quantitatively also from Fig. 2?

The Conclusions section, as noted before, should emphasize clearly that the method cannot be applied without field data. This is more of a Conclusions than proposing future work, for which the Authors spent one half of the Conclusions section, though necessarily it somehow reduces the relevance of the proposed method.

Figures:

in Fig. 3, it is not clear, at first glance, that the "Surface displacements", "d(D)isplacement sample points" and "SNR > 0.7" and "Elevation differences" legends apply to all the whole figure. Maybe a single legend would be more direct.

in Fig. 4c, maybe a legend would be better than the labels with arrows.

In summary, I believe that the Manuscript should be revised before it can be published as a regular paper on NHESS.

---

## Referee Comment (RC2) · Anonymous Referee #2 · 4 Sep 2017

**Review of the manuscript "Brief communication: 3D landslide motion from cross correlation of UAV-derived morphological attributes"**

submitted by Maria V. Peppa et al.

for the NHESS special Issue "The use of remotely piloted aircraft systems (RPAS) in monitoring applications and management of natural hazards"

**General comments**

This paper describes how data derivatives of repeated UAV flights can be used to investigate the spatio-temporal development of a landslide. The presented workflow builds upon several established methods (SfM-MVS photogrammetry to produce orthophotos and digital elevation models, DEM differencing, image cross correlation to quantify displacements). As a novelty, the study evaluates the use of morphometric features (openness, slope and curvature) instead of (previously used) shaded relief maps to derive displacement magnitudes and directions. Exploring these alternative raster features as input for the cross correlation function is an interesting idea, and I think in general it is suitable for presentation as a "brief communication" in NHESS. For landslide monitoring such a test is relevant because there is the need for improved analysis methods using high resolution data from close-range and remote sensing.

In general, the manuscript is well written and clearly structured. However, there are a couple of points that should be improved (minor revisions) before I can recommend the manuscript for publication.

The most important issue concerns the objective of quantifying 3D motion. I do not completely understand if and how this is achieved. The results of the presented approach are (i) a horizontal (*x-y*) component of landslide motion (2D displacement vectors) and (ii) an elevation difference between two DEMs. These two components can, of course, be jointly used to characterize and interpret the landslide changes. However, I cannot find an explanation how these two datasets are combined to a true 3D data product. Is the elevation difference interpreted as the vertical component (*z*-direction) of landslide motion? The elevation difference at a raster cell is not necessarily caused by a vertical movement; (at least in theory) it could as well result from a purely horizontal displacement of objects/morphological structures. Please consider if this is a valid argument and if this can have implications for interpretation of landslide kinematics. If the presented study actually derived true 3D motions, please provide the required explanations to clarify this. If not, I suggest deleting "3D" from the title to avoid creating expectations by the reader which are not fulfilled by the approach presented in the paper. Instead of "3D motions", the motions may be rather referred to as "horizontal and vertical", or perhaps more appropriate (see above) as "horizontal motions and elevation differences". This should be corrected throughout the manuscript.

Another point that should be improved concerns the limitations identified in the study. A-priori knowledge of the displacement magnitude is required to select suitable parameter values for the tool that in turn derives the displacement magnitude. In areas with strong morphological changes between epochs the cross correlation method fails (which is to be expected). Moreover, surface structures for feature tracking must be identified and marked manually. Finally, spurious displacement vectors are produced and these must be identified and removed manually (so far). With regard

to an automated application of the analysis workflow, these limitations are clearly disadvantageous. The authors address these points properly in the discussion section, but in my opinion, a concise summary of the limitations (at least pointing to their existence) should be included in the conclusions as well.

**Specific comments**

Comment 1 - p. 1 line 13:
What do you mean with "effective"? Consider deleting this word if it has no specific meaning.

Comment 2 - p. 1 lines 16 – 17:
Consider inserting "characteristic" or "new": "the formation of *characteristic/new* surface morphological structures".

Comment 3 - p. 1 line 21:
This is the first use of the abbreviation "UAV" in the main text, so it should be introduced together with its complete name here.

Comment 4 - p. 2 line 6:
Consider replacing "implemented with" by "combined with".

Comment 5 - p. 2 line 29:
How big are the targets you used? The size of the targets could be valuable information for inexperienced colleagues who want to apply the suggested methods for their own monitoring task and benefit from your experiences.

Comment 6 - p. 3 lines 10 - 11:
An estimate for the sensitivity level is important information. However, concerning the approach for obtaining the sensitivity level, "applying error propagation" is quite unspecific. A citation of Peppa et al. (2016) and maybe the respective reference therein (Wolf and Ghilani, 1997) should refer the interested reader to a description of the methodology.

Comment 7 - p. 3 line 20: "using a 3x3 pixel radial distance"
Assuming that you used a square neighbourhood (moving window) of 3x3 pixel, I suggest rephrasing this to "3x3 pixel window".
More importantly, this point raises the question if other neighbourhood sizes have been tested and what impact this had on the results. This, however, is later announced as an objective for future work (in section 7), which seems to be a good idea.

Comment 8 - p. 4 lines 5 - 10:
What exactly is the output from the COSI-Corr function? Two maps, one for displacement in Easting and one for displacement in Northing? Are these components combined? What exactly is shown in Figure 3? The maximum displacement magnitude in any direction or only in Northing (because the slope gradient is directed roughly in N-S direction)?

Comment 9 - p. 4 line 10:
How have these "27 sample points" been selected/defined? Randomly? Or at distinct points (such as rock boulders) that can be visually well detected in successive epochs?

Comment 10 - p. 3 line 13:

This is where suddenly "3D surface deformation" is presented, but it was not mentioned before how this three-dimensionality was achieved (and stored in terms of data structure), since the described cross-correlation analysis and the DEM differencing resulted in "2D motion" and "elevation change" respectively. Are these two components (*x-y* displacement vectors and elevation change) combined to 3D vectors? If yes, how can these 3D movements be analysed and interpreted subsequently?

Comment 11 - p. 3 line 18: "characteristic surface structures were manually located"
Does this mean groups of pixels, which represent these surface structures, were manually selected and extracted from each epoch and then passed as input to the CIAS tool?

Comment 12 - p. 4 line 20:
Can you be more specific please, for which parameters could thresholds be used (for instance)?

Comment 13 - p. 5 line 27: "a threshold of 63°"
This threshold was empirically determined (by try-and-error or systematically by a specific search procedure)?

Comment 14 - p. 10 line 3: Figure 1 caption
Please consider providing the long name for SNR here as well.

Comment 15 - p. 11: Figure 2
Please provide a description for the grey margin (the sensitivity level).
Moreover, (in the text) you could report the number of samples with displacements larger than the sensitivity level (measured by at least one of the two methods or by both methods).

Comment 16 - p. 13: Figure 4 a) and b)
Please include the arrows (displacement vectors from NCC function?) in the legend or caption and maybe you can improve their contrast a bit.
Moreover, it is not clear what the dark red colour is. Does it represent parts where the 0° - 63° classes of both epochs overlap?

---

## Author Comment (AC1) · 17 Oct 2017

**Brief Communication: Landslide motion from cross correlation of UAV-derived morphological attributes**

Maria V. Peppa[1], Jon P. Mills[1], Phil Moore[1], Pauline E. Miller[2], Jonathan E. Chambers[3]

[1]School of Engineering, Newcastle University, Newcastle upon Tyne, UK
[2]The James Hutton Institute, Aberdeen, UK
[3]British Geological Survey, Keyworth Nottingham, UK

*Correspondence to*: Maria V. Peppa (m.v.peppa@ncl.ac.uk)

**Replies to the referees**

We would like to thank the two anonymous referees for their valuable comments and their constructive arguments that have helped improve the manuscript. Replies to the comments received have been addressed separately for each referee below. The line and page numbering used by the referees, which refer to the discussion paper, were also followed here.

Replies to the comments of Referee 1 (R1)

R1 presented a valid argument about the fusion between DEM differencing and image correlation with *openness*. The word "fusion" was erroneously chosen as it misinterpreted the methodological workflow. There is no fusion between the two techniques, a) DEM differencing and b) image correlation. DEM differencing was applied to illustrate the subsequent elevation changes. For instance, the DEM differences indicated the dramatic changes occurred between February and May 2016 (Figure 3f). These elevation changes could explain the NCC function decorrelation (voids in the displacement map, Figure 3c). Additionally, the DEM differences illustrated vegetation growth at the foot of the slope (Figure 3f). Due to vegetation variations, noise was generated (Figure 3c). As was also observed by Referee 2 (R2), elevation differences and displacement maps can be jointly used to interpret landslide deformation. The systematic downward horizontal movement of the eastern lobe is shown in Figures 3a, b, c and Figure 4a. This movement formed ground accumulation, generating positive elevation differences (seen in Figures d, e and f). Figure 4c also illustrates the surface change produced by the horizontal movement along Profile AB. Hence, DEM differencing could support the explanation of errors derived from the image correlation function. It can also illustrate the two types of movement, as observed by R2. Specifically, the first type is the horizontal motion of surface structures (mostly observed over the eastern lobe) and the second type is the vertical change generated by slope failures (as occurred over the western lobe and at the back scarp), in Figures 3e and 3f respectively. To make these points clear, the sentence in the Conclusions (page 7 lines 10-12) "*The analysis has illustrated that the fusion of openness morphological attribute along with DEM differencing can support the comprehensive interpretation of landslide behaviour, providing a holistic overview of 3D surface deformation patterns.*" was changed to: "*The analysis has illustrated that openness implemented with image cross-correlation functions can be used in conjunction with DEM differencing to support the comprehensive interpretation of landslide behaviour, providing a holistic overview of horizontal and vertical deformation patterns.*"

Additionally, R1 commented that DEM differences are hardly mentioned throughout the paper. Numerical results of elevation differences were added in line 20 of page 5 to address this comment. Specifically:

"*Part of the western lobe collapsed, creating a dramatic change of -0.70 m maximum ground loss and a +0.50 m maximum ground accumulation within 11 months (Figure 3e). The surface ruptured at the upper part of the slope, yielding a maximum ground subsidence of approximately -1.70 m and a maximum elevation increase of approximately +1.05 m, as seen in Figure 3f. In addition, Figure 3f depicts the grass growth…*"

Other corrections were added in line 21 of page 5 as below:

*"Also, over the regions with extreme deformations (e.g. back scarp in Figure f), decorrelation created voids on the displacement map (Figure 3c)."*

To demonstrate that DEM differencing supported the landslide interpretation, as also observed by R2, the following sentence was added in line 33, page 6:

*"The episodic surface ruptures generated vertical ground loss and accumulation, as seen in Figures 3e and f. The horizontal downward motion of the front part of the eastern lobe was illustrated as positive elevation change. This motion was also identified with the image cross-correlation analysis (Figure 3)."*

It was suggested that the first paragraph of the Results section be transferred to the Methodology section as it does not represent pure results of the workflow. However, this paragraph constitutes the results of the experiment using synthetic datasets, an essential step to tune the NCC parameters (as correctly characterised by R1, the "calibration" step). To address this comment the following sentence was therefore added to the beginning of the Results section: *"Before presenting the horizontal and vertical displacements over the Hollin Hill landslide, the results of the synthetic experiment are firstly described. All four…"*

Additional changes were made (line 6, page 5):

"*Some scattered points fell within the ±0.10 m 3D sensitivity level shown in grey*, especially for the March-June 2015 and June-September 2015 epoch pairs."

Legends were added to Figure 3 and 4 to aid interpretation (see pages 6 and 7 in the current document).

Replies to the comments of Referee 2 (R2):

In addressing the first comment of R2, the phrases "*3D motion*", "*3D*" and "*3D surface changes*" were removed from the manuscript and replaced by the phrase "*horizontal motions and elevation differences*". To avoid misunderstanding, the word "*3D*" was also removed from the title. This also addresses the specific comment 10 (page 4) regarding the phrase "*3D surface deformation*". To clarify, there was no combination of 3D vectors in the presented work, but a cross-correlation analysis and DEM differencing which produced horizontal 2D motion and elevation changes, respectively.

The second comment concurs with one of the comments of R1, both suggesting that the limitations should be summarised in the Conclusions section, even though they were already mentioned in the Discussion section. This was addressed in line 12, page 7, as follows:

*"Major limitations include the reliance on a priori knowledge of the landslide type and displacement magnitude to tune the image cross-correlation function parameters, use of field data for cross validation, manual surface feature identification and manual cleaning or threshold definition to remove erroneous displacement vectors. These limitations affect the performance of the resulting horizontal motions and elevation changes."*

Answers to specific comments are shown below.

- Comments 1-3, page 1:
    - The word "effective", in line 13, was deleted.
    - The word "characteristic" is added in lines 16-17.
    - The phrase "unmanned aerial vehicles" was added in line 21.
- Comments 4 and 5, page 2:
    - The word "combined" was substituted with the word "implemented".

- In order to add the size of the targets, lines 29 and 1 were changed to:

  *"Circular targets of 0.40 m diameter (equal to 8-10 pixels), with centres easily recognisable in the imagery, were established. Between 11 and 20 targets were surveyed for each of the different campaigns using rapid static Global Navigation Satellite System (GNSS)."*

- Comments 6 and 7, page 3:

  - To improve clarify as to how the sensitivity level was derived the following changes were made in lines 10-11: *"Peppa et al. (2016) described an approach to derive the vertical sensitivity with the use of DEM standard deviations. An approximate ±0.10 m sensitivity level, corresponding to the lowest detectable change, was estimated by applying error propagation (with a 95% confidence level) to the 3D RMSE values, calculated at check points. Both approaches resulted in a sensitivity level of the same order of magnitude."*

  - The phrase "3x3 pixel window" was added in line 20.

- Comments 8, 9, 11 and 12, page 4:

  - COSI-Corr provides displacements in Easting and Northing separately, which can be combined to generate a 2D motion map. To clarify this, the following sentence was added in line 4: *"The computed displacements in Easting and Northing were combined to provide 2D motion maps across successive epochs."* Figure 3 also presents the 2D displacement of this combination. Indeed the displacement magnitude is significantly greater in Northing than Easting.

  - To describe how the 27 sample points were identified, the following sentence was added: *"These points were identified on visually identifiable characteristic surface breaks and evenly distributed across the site with displacement magnitudes from cm- to m-level."*

  - In line 18 the phrase "characteristic surface structures were manually located" means that 2D coordinates from particular positions located on December 14 openness were extracted and used as input to the CIAS tool. To clarify this, the following changes were made: *"Thus, characteristic surface structures were manually located over the December 2014 openness image and the derived 2D coordinates were used as input to the CIAS tool. The planimetric vectors of these locations, between December 2014 and May 2016, were automatically derived with the same tool."*

  - To be more specific for the threshold parameters, the following sentences were added in line 20: *"For instance, the sensitivity level could serve as a threshold to remove vectors of lengths lower than ±0.10 m. Based on previous knowledge of the Hollin Hill landslide (Uhlemann et al., 2017), a specific azimuth range could be used as an additional threshold to exclude vectors showing, for example, backward motion due to rotational failures."* This sentence also partly addresses the comment of R1 regarding the knowledge of presence of local rotational failures (paragraph C2). To clarify further, the following text was added to the Discussion on page 6:

    *"Even though threshold definition can automatically remove spurious vectors, it is not a straightforward process as it relies on a priori knowledge of the landslide. Where such information is unavailable, additional field data may be used. This demonstrates that image cross-correlation performance is strongly related to the landslide movement type. For mixed types, such as the Hollin Hill landslide (a combination of rotational failures with earth flow, as shown in Uhlemann et al. (2017)), the successful application of image cross-correlation is not entirely guaranteed."*

- Comment 13, page 5 line 27:

  - One sentence was added in line 28 to show how the threshold of 63° was derived. *"This threshold was derived with the aid of visual inspection along profiles at multiple locations over active parts of the landslide."*

- Comment 14, page 10 line 3: the full name of SNR was added.

- Comment 15, Figure 2:

- o The number of samples was added in line 6 of page 5: "*33 and 38 sample points across all epoch pairs with displacement magnitude larger than ±0.10 m were observed manually on orthomosaics and automatically derived with COSI-Corr respectively.*"
    - o The gray zone was also added as a legend in Figure 2.
- Comment 16, Figure 4:
    - o Arrows were included in the legend. Their colour was changed from black to blue to improve their contrast.
    - o The dark red region, which represents the 0º-63º class of both epochs openness overlap, was included in the legend.

All changes to Figures 2, 3 and 4 are shown below.

[Figure]

**Figure 2: Scatterplot of estimated surface displacements determined by COSI-Corr with *openness* plotted against manual observation per epoch pair.**

[Figure]

**Figure 3: Maps of surface displacements and elevation differences of (a and d) December 2014 -March 2015, (b and e) March 2015-February 2016 and (c and f) February 2016-May 2016, respectively. Manually derived planimetric vectors at sample points are also superimposed.**

[Figure]

**Figure 4: Detailed view of December 2014 and May 2016 openness maps over (a) eastern lobe and (b) back scarp with elevation and *openness* plotted along (c) Profile AB.**